TOPICAL REVIEW

# The neural control of movement: a century of *in vivo* motor unit recordings is the legacy of Adrian and Bronk

Dario Farina[1] and Simon Gandevia[2]

[1] *Department of Bioengineering, Imperial College London, London, UK*
[2] *Neuroscience Research Australia, Sydney and University of New South Wales, Sydney, Australia*

Handling Editors: Laura Bennet & Mathew Piasecki

The peer review history is available in the Supporting Information section of this article (https://doi.org/10.1113/JP285319#support-information-section).

**Abstract** In two papers dated 1928 to 1929 in *The Journal of Physiology*, Edgar Adrian and Detlev Bronk described recordings from motor nerve and muscle fibres. The recordings from motor nerve fibres required progressive dissection of the nerve until a few fibres remained, from which isolated single fibre activity could be detected. The muscle fibre recordings were performed in humans during voluntary contractions with an intramuscular electrode – the concentric needle electrode – that they describe for the first time in the second paper. They recognised that muscle fibres would respond to each impulse sent by the innervating motor neurone and that therefore muscle fibre recordings provided information on the times of activation of the motor nerve fibres which were as accurate as a direct record from the nerve. These observations and the description of the concentric needle electrode opened the era of motor unit recordings in humans, which have continued for almost a century and have provided a comprehensive view of the neural control of movement at the motor unit level. Despite important advances in technology, many of the

**Dario Farina** has been Full Professor at Aalborg University and at the University Medical Centre Göttingen, where he founded and directed the Institute of Neurorehabilitation Systems, until he moved to Imperial College London as Full Professor and Chair in Neurorehabilitation Engineering. His research focuses on biomedical signal processing, neurorehabilitation technology, and neural control of movement. **Simon Gandevia** is one of the founders of Neuroscience Research Australia in Sydney and he is its Deputy Director. As a clinical neurophysiologist and conjoint Professor at the University of New South Wales, his research has been directed to understanding a range of aspects of the motor and sensory control of human movement including control of respiratory muscles.

The Journal of Physiology

principles of motor unit behaviour that would be investigated in the subsequent decades were canvassed in the two papers by Adrian and Bronk. For example, they described the concomitant motor neurones' recruitment and rate coding for force modulation, synchronisation of motor unit discharges, and the dependence of discharge rate on motor unit recruitment threshold. Here, we summarise their observations and discuss the impact of their work. We highlight the advent of the concentric needle, and its subsequent influence on motor control research.

(Received 18 August 2023; accepted after revision 24 November 2023; first published online 7 December 2023)

**Corresponding author** D. Farina: Department of Bioengineering, Imperial College London, London, UK. Email: d.farina@imperial.ac.uk

**Abstract figure legend** Schematic representation of muscle fibre electrical recordings performed with an intramuscular concentric needle electrode. In healthy conditions, muscle fibres respond to each impulse sent by the innervating motor neurone. The identification of the waveforms of individual motor unit potentials from the selective multiunit recording provides information on the discharge times of the muscle fibre potentials, which are associated to the axonal potential timings.

## Introduction

Edgar Adrian and Sir Charles Sherrington were awarded the Nobel Prize in Physiology or Medicine in 1932 'for their discoveries regarding the functions of neurons.' Both made fundamental discoveries in several areas of physiology and were pioneers in the control of movement. While the recognition to Adrian was associated mainly with his discoveries on neuron functions made with experiments on sensory fibres, here we celebrate the long-lasting impact of his work on motor nerve fibres. Specifically, we discuss the studies he performed with Detlev Bronk, an American biophysicist who obtained a National Research Council fellowship to study at Cambridge with Adrian and who would later contribute substantially to the study of cardiovascular regulation.[1]

Influenced by the work of Santiago Ramón y Cajal on the fundamentals of the connectivity and function of the nervous system (1904), Liddell and Sherrington (1925) introduced the concept of the motor unit as one motor neurone that innervates a group of muscle fibres. Shortly after, a pair of papers by Adrian and Bronk in *The Journal of Physiology* (Adrian & Bronk, 1928; 1929) described the output of the spinal cord during autonomous (respiration), reflex and voluntary contractions by direct and indirect motor nerve fibre recordings in animals and humans. In the second paper, the authors unveiled a technological advance for recording from muscles to obtain information on motor neurone discharges comparable to that gathered from direct nerve fibre recordings. They recognised the link between electrical activity in the motor nerve fibres, which required a complex nerve dissection in animals for a direct recording, and recordings from muscle fibres. They argued that (Adrian & Bronk, 1929; p. 132) "muscle fibres should certainly respond to every impulse reaching them from the nerve. Consequently, the electric responses in the individual muscle fibres should give just as accurate a measure of the nerve fibre frequency as the record made from the nerve itself." This statement links motor axonal action potentials to muscle fibre action potentials, and reveals that we can record the activity of spinal motor neurones 'remotely' by measurement of the electrical activity of the groups of muscle fibres they innervate. We will refer to the group of muscle fibres innervated by a single motor neurone as the 'muscle unit').[2]

In those years, Adrian and Bronk were among the first to recognise the association between axonal and muscle unit potentials and its implications. At the same time and partly independently, Denny-Brown (1928, 1929) was recording motor unit potentials in animal preparations from isolated muscles during reflex contractions, with an

---

[1] After his return from England in the late 1930's, Bronk held positions at the University of Pennsylvania, Cornell, Johns Hopkins University, and the Rockefeller Institute, progressively moving towards administrative and management roles. In his later career, Bronk had a substantial influence on science policy at the national level, having held executive positions at the National Academy of Sciences and the National Research Council. These efforts culminated with his election to the presidency of the National Academy of Sciences in 1950, which allowed him to influence national science policy for more than a decade.

[2] The relation between axonal and muscle fibre action potentials was *a priori* assumed as natural by Adrian and Bronk. Indeed, also at present, the neuromuscular junction is considered a very safe synapse that always results in the generation of a muscle fibre potential when reached by an axonal potential. While this can be argued indirectly when stimulating the nerve branches with transcutaneous stimulation or individual axons with microneurographic methods (e.g., Thomas et al., 1990; Westling et al., 1990), a direct experimental validation during voluntary contractions is complex. Moreover, while the safety margin of the neuromuscular junction is likely to be large in healthy, young individuals, it can decline with pathology (e.g. Drost et al., 2001).

electromyographic (EMG) system developed some years earlier by Sherrington (1921) for global EMG recordings. Moreover, in the same period in Germany, Wachholder (1928) was using needle electrodes of various materials, previously developed by Rehn (1921), for human muscle recordings in a similar way as Adrian and Bronk used their concentric needle.[3] Nonetheless, Adrian and Bronk recognised the simplicity of the recording from muscles during voluntary contractions, the association with motor neurone activities, and the fact that the approach would have allowed them to compare data from animal to human studies.

The electrode that Adrian and Bronk proposed to record from muscle tissue effectively opened the era of *in vivo* motor unit investigations in humans. They noticed that recordings from muscles were simpler than those from nerves, which required dissection in their experiments, and could be done in the *intact* muscle during *voluntary* contractions (Adrian & Bronk, 1929; p. 132): "We have not attempted to dissect out individual muscle fibres, or groups supplied by a single nerve fibre, since most of the information we need can be readily obtained by the use of very small electrodes buried in

the substance of the intact muscle." A small electrode was sufficient to separate the activity elicited by individual motor nerve fibres in the muscle tissue. They describe this electrode – the concentric needle electrode – as (Adrian & Bronk, 1929; p. 133) "An enamelled copper wire, No. 36 gauge, i.e. 193 $\mu$m diam., [that] is passed up the centre of a small hypodermic steel needle and held in position by a plug." (Fig. 1). Not only is this design still used after almost a century from this original description, but over the decades it also inspired progressively more sophisticated electrode designs.

To celebrate the two papers of Adrian and Bronk, we first provide an overview of their original observations, which anticipate many physiological mechanisms that would be further studied in subsequent years. Further, we discuss the implications of these observations as well as their proposal to study nerve, and therefore motor neurone, activity by recording signals from human muscles. This proposal provided a novel window into the neural control of movement and the potential for translation of physiological investigations in animals to human studies. Finally, we briefly discuss the development of the field following these two influential papers, specifically identifying discoveries of mechanism of motor neurone activity after the two papers as well as the development of clinical EMG.

---

[3]The needle electrodes used by Wachholder (1928) and previously by Rehn were made of platinum, or steel (sewing needles), or nickel silver and measured activity quite selectively in muscles. In some of the figures by Wachholder it is also possible to observe potentials that may belong to individual motor units (e.g. his Fig. 10). Yet, Wachholder did not interpret them as single motor unit activity and provided an explanation of these activities as a mixture of interference patterns and waveforms changing amplitude over time (see his explanatory Fig. 11).

## Physiological insights by Adrian and Bronk

Much of the two papers describes observations of the discharge characteristics of the motor nerve fibres by

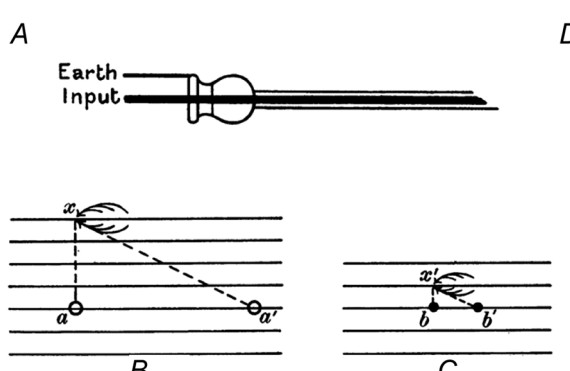
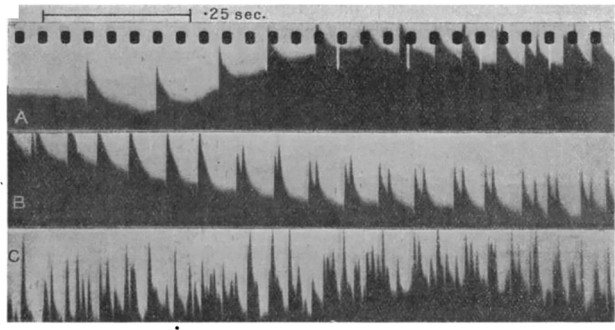

**Figure 1. The concentric needle electrode as originally described by Adrian and Bronk**
*A*, the cannula of the needle is the reference and an electrode inside it records the signal with respect to this reference. *B* and *C*, original drawings by Adrian and Bronk schematically showing the electrode selectivity. Bipolar detection electrodes far away from each other, as in *B* in the positions *a* and *a'*, would record the activity from many and distant muscle fibres, such as located in the position *x*, whereas closer electrodes, as in *C* in the positions *b* and *b'*, would have greater selectivity and record the activity of muscle fibres in the vicinity only, such as in the location marked as *x'*. *D*, original recordings with the concentric needle from the human triceps muscle. The rows (A, B, C on the traces) correspond to progressively greater levels of force. Isolated motor unit discharges repeat at a rate of approximately 10 pulses per second at the start of the contraction (top row). The discharge frequency doubles when force increases (middle row). Action potentials from other motor units are visible after this point, with a multi-unit signal in the bottom row. Adapted and reproduced with permission from Adrian & Bronk (1929).

direct nerve recordings. Recordings of the activity of single nerve fibres were performed with a dissection technique described in the first paper, where it was mainly used in rabbits anaesthetised with urethane. The method required progressively splitting the nerve longitudinally under a microscope so as to leave as few undamaged fibres as possible in the middle in continuity. Finally, the remaining fibres were separated from each other until the two ends of the nerve were joined with a small narrow strand (Fig. 2). Recordings from this strand did not usually contain activity from a single fibre but from a sufficiently small number of fibres to be able to separate their activities from the global electrical recording from the dissected nerve. This elaborate experimental method could be applied only in anaesthetised animals, and therefore the first paper was dedicated to autonomous activity, with recordings mainly from the phrenic nerve.

When recording from the dissected phrenic nerve, they first observed the repetition of action potentials with approximately constant amplitude ('all-or-nothing' response). They noted that the repetition of action potentials of the same shape and size, with frequency modulation, mirrored the observation Adrian made a few years earlier on sensory fibres, where the intensity of the sensory stimulus was coded in the frequency of repetition of identical impulses (Adrian, 1926a, b; Adrian & Zotterman, 1926a, b). As we noted above, it is thanks to

these earlier observations on the nature of sensory nerve action potentials and on the neural basis of sensation, that some years later, in 1932, Adrian shared the Nobel Prize for physiology or medicine with Sherrington for their contributions towards our understanding of the function of neurons.

The discharge frequencies that Adrian and Bronk observed for the action potentials of motor fibres in the phrenic nerve were as low as 20 pulses per second (Fig. 2*C*). These frequencies appeared to them strangely low as it was known that they would not cause a tetanic contraction of the muscle fibres. Nonetheless, they correctly recognised that low discharge rates for individual motor nerve fibres could still produce smooth muscle contractions because of the asynchronous activation of different nerve fibres (spatial asynchronous activation). Interestingly, similar reasoning was used by Denny-Brown (1929), who in 1929 described the properties of single motor unit potentials in the decerebrate cat during the stretch reflex and who would then go on to lay the foundations of clinical EMG in the 1930s to 1940s, which we will discuss later. Adrian and Bronk received the paper by Denny-Brown when finalising the last version of their papers and included a note commenting on his results.

One of the most interesting findings of the first paper concerns how force increases, a topic that would be further discussed in the second paper. They first showed

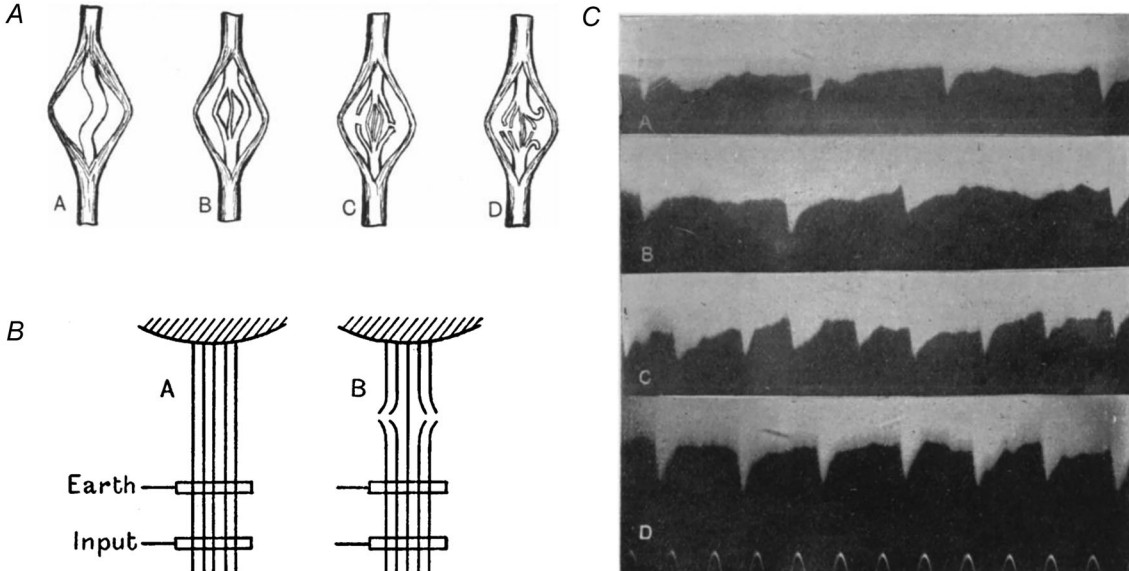

**Figure 2. Nerve dissection technique**
*A*, the nerve is progressively dissected longitudinally until only a few fibres remain in continuity. *B*, the dissection and recording techniques are schematically represented (A, before dissection, B after dissection). *C*, records from the 3rd cervical root of the phrenic nerve after dissection. Nerve fibre potentials are represented as downward deflections of the white signal. A, B: normal breathing with average discharge of approximately 27 pulses per second. C, D: occlusion of air tubes determines an increase in discharge frequency to approximately 68 (C) and 55 (D) pulses per second. D reports at the bottom the peaks of an oscillatory signal used for time reference. The estimated time interval between successive peaks is 10 ms. Adapted and reproduced with permission from Adrian & Bronk (1928).

that the range of frequencies of the single fibres (20 pulses per second to 50 to 80 pulses per second; Fig. 2) corresponded to the maximum slope for the change in force as a function of the discharge frequency, i.e. a change in discharge frequency in the measured range corresponded to the largest corresponding change in force. Further, they concluded that modulation in firing rate alone would have not been sufficient to explain the force required by the muscle in normal breathing and in conditions when the air supply was cut off. They observed that (Adrian & Bronk, 1928; p. 94) "… the force of contraction must increase many times when the air tubes are clamped, whereas an increase in the discharge frequency from 20 to 60 a sec would not be likely to account for much more than a two-fold increase." (see also Fig. 2). Further reasoning that supported the presence of progressive motor neurone recruitment in addition to firing modulation, related to the time span over which single fibre discharges were observed as compared to the duration of the whole nerve discharges: single fibre recordings did not have the same duration as the full nerve recordings, in agreement with the presence of some form of recruitment. This was a crude but appropriate consideration. Overall, they recognised the need for a mechanism of progressive recruitment of motor neurones, as contrasted with the "d'emblée" activation (all together), which was an alternative mechanism. These mechanisms, now well established, were still extensively debated. For example, when discussing the stretch reflex in the decerebrate cat, Denny-Brown (1929) observed that each unit was suddenly recruited and then maintained an approximately constant firing rate despite a continuous increase in tension. Denny-Brown (1929; p. 270) commented: "Recruitment of the units is a constant finding, and increase of rate, to the extent found by Adrian and Bronk (2) in the phrenic discharge, does not here occur".

Interestingly, the recordings of motor nerve fibre potentials from the phrenic nerve in the rabbit by Adrian and Bronk were confirmed in humans only 70 years later. Needle recordings from the human diaphragm during breathing (while the diaphragm is shortening and lengthening) showed that force in this muscle is indeed increased by both recruitment of new motor neurones and increasing their rate of discharge (Butler et al., 1999; De Troyer et al., 1997).

In the first paper, Adrian and Bronk also speculated about synchronisation of discharges of motor nerve fibres. They assumed that the degree of synchronisation would be low at low forces when the nerve fibres discharged at low rates but high when the nerve fibres discharged at maximal rates. The reasoning closely followed the discussion explaining the observed low firing rates. Along this line, a smooth production of muscle force would occur anyway at high discharge rates, even with full synchronisation,

since the firing rates were supposed to be high and contractions of each unit tetanic. They stated (Adrian & Bronk, 1928; p. 96): "There is no doubt that the chief condition for the synchronous activity of the neurones is that the discharge should be near its maximum intensity, but we have not been able to make out what other conditions may favour or hinder its appearance." This reasoning on synchronisation, of which validity they were themselves cautious, is one of few speculations that has not been confirmed subsequently, as we will discuss later. It is also interesting that Adrian and Bronk probably assumed that all motor units would discharge at maximum firing rates during very strong contractions, so that they would all produce tetanic contractions. Conversely, neither of these two assumptions (maximal rates and tetanic contraction for all units) is correct. Indeed, high threshold motor unis are recruited after and usually achieve lower discharge rates than lower threshold ones (De Luca et al., 1982) and maximal tetanic forces are reached at activation rates as high as 100 Hz, generally greater than those achieved during voluntary contractions (e.g. Macefield et al., 1996).

The second paper focuses mainly on reflex and voluntary contractions. Here they chose several nerves for dissection, mainly in the hindlimb of the cat. In these experiments, the muscle afferents were cut. While recording in various conditions, such as when pressing or pinching the foot of the cat, they observed recruitment and derecruitment curves for single motor nerve fibres (their Fig. 5 in the second paper), confirming the conclusions of the first paper on recruitment and rate coding. These curves showed all the main characteristics that would be reported over later decades, such as a greater discharge rate at recruitment than at derecruitment. They also observed a large range of discharge rates from as low as 5 to as high as 90 pulses per second, and they critically observed that the degree of recruitment can differ between muscles (partly explaining contradictory results discussed above by Denny-Brown (1929)), being more prominent in some muscles than others.

Then, they introduced the concentric needle and described the first motor unit recordings in humans during voluntary contractions, mainly targeting the triceps muscle (see Fig. 1*D*). With this approach, they identified discharge rates as low as 6 pulses per second at very low forces. The observed action potentials were interpreted as activity from either single muscle fibres or small groups of fibres innervated by a single axon. Determination of the number of fibres contributing to a recorded intramuscular potential would have been complex. This was exemplified, years later, by the amount of work and evidence that Stålberg and collaborators had to provide to convince the scientific community that the action potentials they recorded with their multielectrode needle were single fibre recordings (Ekstedt et al., 1969). Adrian and Bronk recognised that they could not

determine the exact portion of the muscle unit they were recording from with their electrode but, for the purpose of their investigations, this was irrelevant as they only needed activity that mirrored the axonal action potentials.

They further noticed that the EMG signal recorded with the concentric needle became more complex as force increased (this is evident in the bottom row of Fig. 1*D*) but also observed that it was still possible to separate at least two waveforms from the recordings, i.e. the activity of at least two motor units. This differentiation of the activity of different motor units was based on the waveforms of the potentials, which was also common in other animal studies (Denny-Brown, 1928, 1929). This approach was a premonition about modern techniques of separation of action potential trains, which we will discuss later.

In agreement with the results obtained during breathing in the first paper, Adrian and Bronk identified concurrent recruitment and rate coding for force production in humans during voluntary contractions (Adrian & Bronk, 1929; p. 135): "Records of this kind show that in each group of muscle fibres the frequency of excitation may be so low that there can be very little summation of the contractile effect; that the frequency rises as the contraction becomes more powerful, and that at the same time more and more fibre groups come into play." They further state even more clearly (Adrian & Bronk, 1929; p. 137): "We conclude that the voluntary contraction in man is maintained, like the reflex contractions in the cat, by a series of nerve impulses which range from 5 to 50 or more a sec. in each nerve fibre, and that the gradation in force is brought about by changes in the discharge frequency in each fibre and also by changes in the number of fibres in action."[4]

Their Fig. 21 in the second paper (reproduced here in Fig. 3*B*) is a sketch that includes rate coding and recruitment as well as the notion that motor units recruited earlier would achieve higher firing rates than later recruited units at a given force level [this mechanism, although still debated, was later named the 'onion skin' phenomenon of motor unit recruitment (De Luca & Erim, 1994)]. In their scheme, the missing aspects are the orderly recruitment by size (which we will discuss later) and the concept of common synaptic input received by groups of motor neurones. Without these two concepts, Adrian and Bronk were forced to assume a partly non-uniform input to motor neurones (see Fig. 3 and the discussion we provide later on it) to explain progressive recruitment and the range of discharge frequencies among motor neurones.

Overall, the two papers are packed with physiological insights, with a precise identification of several mechanisms of motor unit control and accurate insight into the match between such control and the characteristics of force production by muscle fibres. The identified mechanisms have been confirmed repeatedly over the years, with progressively more advanced methods for motor unit recording and analysis.

## Evolution of methods for motor unit investigations *in vivo*

After the concentric needle, several other electrode systems have been developed, including intramuscular/epimysial wire electrodes and multi-channel intramuscular electrodes. In parallel there have been advances in the automatic algorithms for the identification of potentials from multi-unit recordings and efforts towards the non-invasive analysis of motor units.

Electrode developments following the concentric needle include the bipolar wire electrode (Basmajian & Stecko, 1962; Bigland & Lippold, 1954), the multielectrode needle by Buchthal et al. (1955), which led to the miniaturised multielectrode needle for single fibre recordings by Stålberg (Ekstedt et al., 1969), tungsten microelectrodes originally proposed for microneurography (Hagbarth & Vallbo, 1969) and later used for muscle fibre recordings (Bigland-Ritchie et al., 1983), the quadrifilar needle electrode (De Luca & Forrest, 1972), the macro-EMG electrode (Stålberg, 1980), the branched electrode (Gydikov et al., 1986), and recent microfabricated wire arrays of intramuscular electrodes (Chung et al., 2023; Muceli et al., 2022). Automatic and high-precision decomposition algorithms originated in the 1970s with the work of De Luca and collaborators (LeFever & De Luca, 1978) and have been further developed for the subsequent 50 years (e.g. LeFever & De Luca, 1982; LeFever et al., 1982; Mambrito & De Luca, 1984; McGill & Dorfman, 1985; McGill et al., 1985; Ren et al., 2018; Stashuk & de Bruin, 1988; Stashuk & Naphan, 1992).

Further parallel developments in electrodes and decomposition algorithms have also allowed the detection of single motor unit activity non-invasively from the skin surface (surface EMG), still based on the fundamental assumption that muscle fibre potentials correspond to axonal potentials. For example, knowledge about circuitry within the spinal cord has greatly progressed by studying the discharge times of motor units recorded

---

[4]As said above, the variability of these mechanism and relatively importance of recruitment and rate coding across different muscles was recognised by Adrian and Bronk. Yet, differences across muscles are probably greater than they thought. For example, it is perhaps fortunate that they did not make the first human recordings from the genioglossus muscle (Adrian used his triceps brachii instead): the genioglossus has motor neurones that fire tonically throughout the breath or phasically in either inspiration or expiration (or in combinations of these patterns; see Saboisky et al., 2006). These patterns and hence the timing and distribution of neural drive differ among the synergist inspiratory muscles (Butler & Gandevia, 2008).

from superficial limb muscles with surface electrodes (e.g. Burke et al., 1994; Marchand-Pauvert et al., 1999; Pierrot-Deseilligny & Burke, 2012). Motor unit recordings with surface arrays of several electrodes have also been reported since the 1980s (e.g. Masuda et al., 1985). These developments have progressively allowed for the non-invasive decoding of voluntary motor neurone activity in superficial muscles of humans (for a recent review, see Farina & Enoka, 2023).

Modern methods for the recording and processing of myoelectric signals in humans with the aim to detect activity of individual motor units remain firmly founded on the concepts and techniques described by Adrian and Bronk. The electrodes have been improved in terms of number of channels and fabrication methods, but they record selective signals from muscles, which was the core idea of Adrian and Bronk (Fig. 1*B* and *C*). The processing methods are now automatic algorithms or analogue real-time systems that replace visual inspection, but they are based on the discrimination of action potential waveforms of different motor units, as Adrian and Bronk could do for a few motor units. Alternatively, in recent decades there have also been developments on

quantitative EMG, based on the analysis of the interference pattern without direct discrimination of motor unit potentials (e.g. Nandedkar et al., 1986a, b). Yet, in most cases, these methods attempt to indirectly assess motor unit properties or behaviour from the interference EMG pattern analysis. The foundation set by Adrian and Bronk has been built on for almost a century of engineering developments of modern techniques for clinical and scientific use of myoelectric signals to study the final output of the nervous system in humans.

## Modern insights into THE neural control of movement

Since the methods pioneered by Adrian and Bronk, scientists could study motor units *in vivo* in humans. We have learned much about how motor units are controlled and the inputs they receive from muscle and other afferents as well as supraspinal centres, in healthy conditions, exercise training, and pathology. Yet, as we have discussed, many of these principles were already enunciated or at least anticipated in the papers by Adrian and Bronk (Fig. 3). Therefore, what have we learned following their two landmark papers?

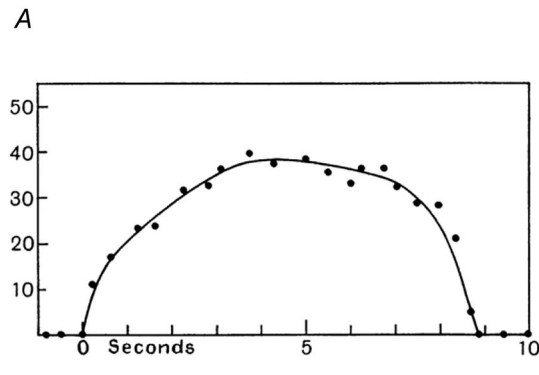

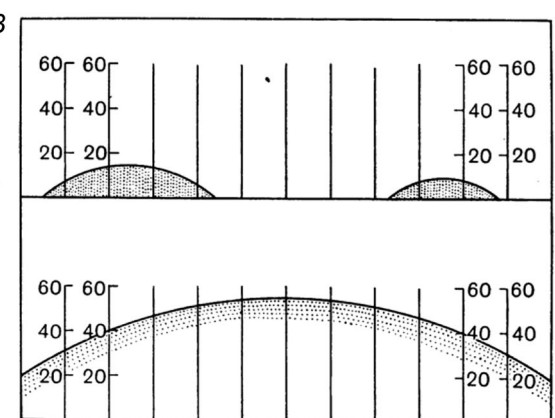

**Figure 3. Recruitment and rate coding**
*A*, the discharge rate (vertical axis) of impulses recorded from the nerve fibres innervating the tibialis anterior during flexion reflex of the cat. Rate coding is evident, as well as the minimal discharge rate at recruitment and de-recruitment. *B*, schematic interpretation of the data in *A* and concluding scheme of excitation and recruitment provided by Adrian and Bronk. In this case, the motor units are represented along the *x*-axis with a vertical line (therefore the *x*-axis here does not represent time, as in *A*, but rather the distribution of motor units). The height along the vertical lines for each motor unit represents the firing rate. The curved lines (two 'bumps' at the top and one larger circular line at the bottom) represent excitation so that when these lines intersect each vertical line, the corresponding motor unit is active and discharges at the firing rate indicated by the height of the intersection along the vertical line (discharge rate increases along the vertical lines). At low forces, excitation is scattered among motor neurones (the two 'bumps' in the top panel) (note that we now know that this is not the case as motor neurones in a pool receive mainly common excitation and are ordered by size in terms of recruitment) and only a few of them are active, discharging at low frequencies. When the force increases, excitation begins to activate most or all motor neurones (in the lower panel, the excitation curve intersects all the vertical lines reported in this plot). To explain a distribution of discharge rates at a given force level, as opposed to all motor units having the same discharge rate, the distribution of excitation is different across motor neurones. This scheme is accurate in terms of the basic mechanisms of force increase and net excitatory input to motor neurones, with the main exception of the orderly recruitment of motor neurones, which would be observed shortly after and formally enunciated a few decades later. Orderly recruitment by size would explain progressive recruitment without the need for differences in net excitatory inputs to motor neurones. Reproduced with permission from Adrian & Bronk (1929).

On inspection of Fig. 3, the distribution of net excitatory input is assumed to be different among motor neurones innervating a muscle. For example, in the top part of panel (b), which represents a low force contraction, excitation is only projected to some and not all motor neurones (see the two distinct curves on the left and right). Adrian and Bronk needed this assumption of selective excitation of motor neurones to explain progressive recruitment and the range of firing frequencies across motor neurones. The same input to all motor neurones in their scheme would have led to all active motor neurones having the same firing rate. They lacked the concept of a large range of excitation thresholds among motor neurones and their distribution according to motor neurone size. A distribution of excitability among motor neurones would have explained the range of firing frequencies and progressive recruitment without the need to assume selective projection of excitatory input to different motor neurones (for a later scheme of recruitment and rate coding that explains a distribution of firing rates and progressive recruitment with the same excitation to all motor neurones, the reader may refer to, e.g. Fig. 4 in De Luca & Erim, 1994).

The distribution of recruitment thresholds across a motor neurone pool is probably the most important physiological discovery about the control of motor neurones following the work of Adrian and Bronk. Motor neurones have a threshold of activation that is linked to their size, so that smaller neurones are activated before larger ones when receiving the same net excitatory input (so called orderly recruitment).

The fact that motor neurones were of a variety of sizes in the same pool and had different activation thresholds was known when Adrian and Bronk published their papers (Sherrington, 1929). Some years later, Denny-Brown, who was a student of Sherrington, and Pennybacker (1938) reported needle EMG recordings during a voluntary contraction of the biceps brachii muscle in humans, stating (p. 324) "This 'recruitment' of motor units into willed contraction is identical with that occurring in certain reflexes … The early motor units in normal gradual voluntary contraction are always in our experience small ones (Fig. 11). The larger and more powerful motor units, each controlling many more muscle fibres, enter contraction late." While these observations clearly revealed a recruitment order by size, they were not yet substantiated by a clear identification of the neural mechanisms underlying the observed orderly recruitment.

Thirty years after the papers by Adrian and Bronk, orderly recruitment was clearly enunciated as a size principle, potentially extending to all neurons, by Elwood Henneman (1957) (Fig. 4), who also explained the biophysical and neural basis of this mechanism in subsequent work (e.g. Henneman et al., 1965). Based also on early work by Katz and Thesleff (1957), Henneman recognised that (Henneman et al. 1965; p. 574) "an equal degree of presynaptic excitation may generate a larger synaptic potential in small cells due simply to their dimensions," and that the net excitatory input required for the activation of a motor neurone is directly associated to the motor neurone size. Because of a broadly distributed input to the motor neurones, the activation of motor neurones should proceed from small to large auto-

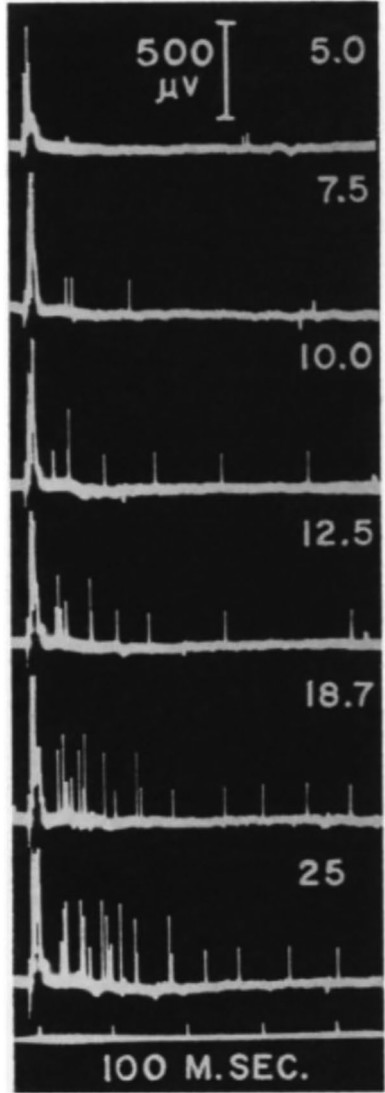

**Figure 4. Henneman's size principle**
Experiments conducted in cats with transected spinal cord below the obex. Electrical stimulation of dorsal nerve roots was used to elicit reflexes in the lumbar ventral roots. The stimulation provoked a large compound potential at the beginning of each recording, followed by asynchronous motor unit activity. The recordings were made from fine filaments of ventral roots dissected to isolate a small number of action potential trains. A progressive increase in the stimulation current recruited more motor units and their size increased. Reproduced with permission from Henneman (1957).

matically and, because of the association between size of the motor neurone, axonal size, and number of innervated fibres, this corresponds to an order from weak to strong motor units. These associations are commonly known as the size principle, or Henneman's size principle[5] (Fig. 4). Following its enunciation in the 1950s, this principle has been tested and challenged in a variety of conditions but is still strongly supported by solid evidence (e.g. Hug et al., 2023).

A further mechanism that was not identified fully by Adrian and Bronk is the correlation of the activities of motor units. They discussed synchronisation in their papers but did not recognise that the tendency of motor unit firings to synchronise should be due to correlated input to the motor neurones. Therefore, their comments on synchronisation remained somewhat speculative and strongly linked to force generation. For example, as discussed above, they speculated that synchronisation could not occur at low contraction forces since at low forces most motor units will generate unfused force twitches and smooth force generation would require highly uncorrelated discharges. Conversely, they argued that synchronisation would occur at high forces where the twitches would be mostly fully fused into a tetanic contraction. We now know that these considerations are not accurate for many reasons.

Correlation between motor unit activities has been studied with several mathematical approaches and defined

with different terms that captured part of this mechanism (Fig. 5). The computation of common drive (De Luca, 1985; De Luca & Erim, 1994; De Luca et al., 1982), short-term synchronisation (Datta & Stephens, 1990; Davey et al., 1993; De Luca et al., 1993; Kozhina et al., 1991; Nordstrom et al., 1992; Sears & Stagg, 1976), coherence functions in the frequency domain (Elble & Randall, 1976; Farmer et al., 1993) between motor unit discharge time series are examples of estimates of correlation between pairs of motor unit activities. They all point to the presence of a common input received by pools of motor neurones. After all, common input is a necessary condition for force modulation. If force needs to be increased (or decreased), a control signal projected to all motor neurones active for the task will need to increase (or decrease). Conversely, if there were multiple inputs to motor neurones going in opposite directions, the effects would partly compensate and modulation of force would be unnecessarily complex. The fact that excitation to motor neurones should proceed in the same direction for the entire pool can also be seen from the plot of Adrian and Bronk (here Fig. 3*B*). By comparing the top and bottom part of Fig. 3*B*, we observe that the curves representing the excitation of motor neurones intersect more and more motor neurones (recruitment) and cross the vertical lines representing firing rate at greater values progressively (rate coding). For example, the excitation curves in the top part of the panel intersect the vertical lines that represent the firing rate at values lower than 20 pulses per second. Conversely, in the bottom part of the panel, excitation has increased and it intersects the firing rate lines at values greater than 20 pulses per second and up to >50 pulses per second. Thus, when force increases, the excitation increases for all neurones in that plot, which is necessary to explain why all motor neurons increase their firing rate when force increases. However, in their original plot, Adrian and Bronk needed the input to be different (although its trend over time was the same) across motor neurones.

We can now group all the classic approaches for assessing correlation between motor neurone output within a single explanatory model of motor neurones receiving a certain portion of common synaptic input over a broad frequency range that inevitably imposes a correlation in the motor neurone outputs. The size principle implies that even with the same input projected to a pool of neurones, the recruitment will proceed progressive from small to large neurones and, consequently, there will be a distribution of firing rates over a wide range. Because of the need for a common input to modulate force, synchronisation between firings of motor neurones is a necessary observation as common (correlated) input to motor neurones necessarily generates some level of correlation in the motor neurone output. In this view, synchronisation, at least as classically defined (Fig. 5), has limited functional significance and

---

[5]Because of the cited early observations by Denny-Brown on an orderly recruitment by size, some have proposed that the size principle should be renamed after him and Pennybacker instead of after Henneman (Vilensky & Gilman, 1998). It is also often noted that Henneman did not acknowledge sufficiently Denny-Brown's contribution in the determination of the size principle (for example, the landmark paper by Denny-Brown and Pennybacker of 1938 is not cited in Henneman's paper of 1957). While these considerations are correct, the fundamental ideas of a distributed excitatory input to motor neurones with an effect directly associated to the motor neurone size were developed by Henneman (see also commentary by Fuglevand, 1998, in response to Vilensky & Gilman, 1998). Moreover, Henneman recognised in his 1957 paper that the size principle would likely have extended to any populations of neurones, as stated in his concluding remark (p. 1346): "… the hypothesis may be advanced for consideration that throughout the nervous system the susceptibility of neurons to discharge is a function of their size." Henneman also provided a biophysical explanation of the size principle in his subsequent work, in particular in the cited paper of 1965 and subsequent ones. For this, he correctly interpreted previous results by Katz and Thesleff (1957) on muscle fibres of the frog that showed that the size of a cell (muscle fibres in the case of Katz and Thesleff) would determine "the effectiveness of current passing through the membrane in bringing about depolarization" (Henneman et al., 1965; p. 575). It is our opinion that the enunciation of a well-defined principle, its underlying biophysical and neural mechanisms, and its generalisation to other types of neurons, should be mostly recognised as due to the work of Henneman.

is rather the by-product of the necessary correlation of the inputs to many motor neurones (Negro & Farina, 2011, 2012). Of note is that the relatively low levels of synchronisation estimated from pairs of motor units with classic approaches (e.g. Nordstrom et al., 1992) do not necessarily indicate that the level of common input is weak. This is due to both the presence of independent input to motor neurones and to the non-linearity of motor neurones in the transmission of input signals to their output (De la Rocha et al., 2007). When a population analysis is performed, rather than a pair-wise analysis of motor units, independent inputs and non-linearities have a smaller impact and common input can be identified as a strong component of the inputs received by motor neurones (Farina & Negro, 2015; Farina et al., 2014; 2016).

Interestingly, it is now technically possible to identify the activities of a very large number of motor neurones in vivo in humans (e.g. Muceli et al., 2022), from which we can observe multiple groups of motor neurones, each receiving different common inputs. The activation of these groups by the central nervous system may represent a modular organisation of movement, as an extension of the classic muscle synergy theory (Bernstein, 1967), to a finer scale of analysis (Hug et al., 2023).

Other physiological insights after the papers by Adrian and Bronk relate to neuromodulatory pathways to motor neurones which alter the motor neurones' excitability. They change the response of motor neurones to a certain level of excitatory drive (Conway et al., 1988; Eken et al. 1989; Hounsgaard & Kiehn, 1989; Hounsgaard et al., 1984). For example, motor unit firing rates have been observed in some conditions to 'saturate,' i.e. they level off even when their drive increases. This mechanism is probably explained by changes in intrinsic excitability of motor neurones (Fuglevand et al., 2015) and may differ for motor neurones of different sizes(Grande et al., 2007). For this reason, saturation may disrupt the earlier proposed 'onion skin' distribution of firing rates.

We also have extended our knowledge substantially with the study of motor unit activity in specific physiological conditions, such as fatigue (Bigland-Ritchie, 1981, 1984; Bigland-Ritchie et al., 1983; Gandevia, 2001), following exercise training (Christie & Kamen, 2010; Del Vecchio et al., 2019; Duchateau et al., 2005; Van Cutsem et al., 1998), and with ageing (Galganski et al., 1993; Gutmann & Hanzlíková, 1996; Nelson et al., 1984; Spiegel et al., 1996). In addition, our knowledge about spinal cord circuitries has advanced dramatically by studying the discharge times of single motor units (e.g. Türker & Cheng, 1994; Türker & Powers, 2005; Pierrot-Deseilligny & Burke, 2012). In these fields of investigation, which we do not review here, the motor neurone has been a window into the working principles of the central nervous system and its analysis has clarified important properties and adaptations of the system.

While corroborating its role as a primary tool for physiological investigations, the technique of motor unit recordings evolved in parallel as a method for clinical analyses. The concentric needle is still used in clinical neurophysiology examinations in medical centres globally to diagnose neuromuscular disorders, such as amyotrophic lateral sclerosis (ALS). Diagnosis

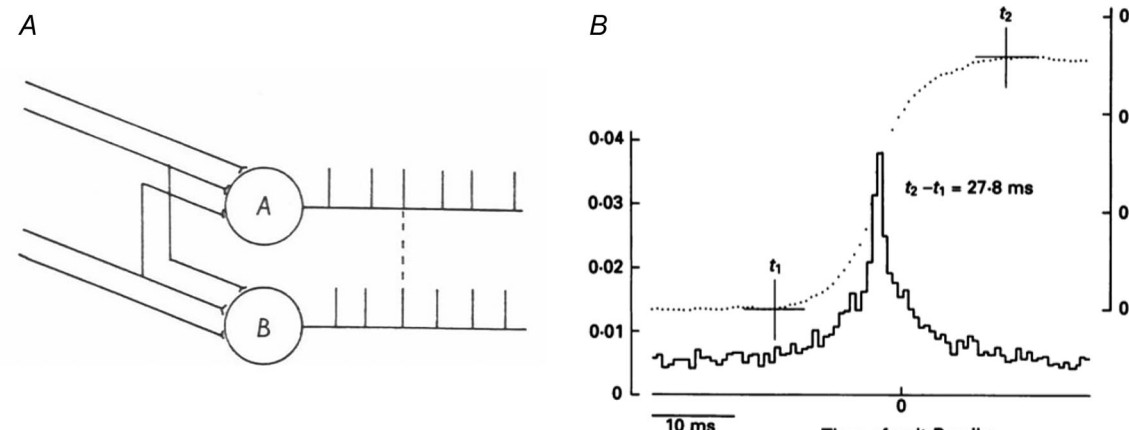

**Figure 5. Motor unit synchronisation**
*A*, when recording the discharge times of pairs of motor neurones, there is a probability above chance level that some discharges occur at very close time instants. This can be quantified by cross-histograms, as those reported in *B*, between the two series of discharge times. In this example, two motor units of the human first dorsal interosseous muscle are analysed. The subject was required to maintain a voluntary contraction so that the two units fired steadily and concurrently, with one, unit A, firing at approximately 10 impulses per second. The firings of motor unit A were found 109% more often in association with those of motor unit B spikes than would have been expected were the pair of units firing independently. Bin width 0–64 ms, 8192 sweeps. Reproduced with permission from Sears & Stagg (1976) (*A*), and Datta & Stephens (1990) (*B*).

by EMG with needle electrodes currently remains by far the most relevant application among the (invasive and non-invasive) EMG techniques.

The most influential pioneering result on the application of the concentric needle in patients has probably been the distinction of action potentials of single denervated muscle fibres, termed fibrillations, from the action potentials of discharging single motor neurons, termed fasciculations, by Denny-Brown and Pennybacker (1938). This allowed separation of muscle weakness due to neurogenic causes, such as ALS and traumatic nerve section, from myogenic causes, such as muscular dystrophy. The report by Denny-Brown and Pennybacker (1938) showed for the first time a clear clinical use of the concentric needle of Adrian and Bronk and marked the beginning of clinical EMG, i.e. the association between EMG findings and likely pathology.

Following the report by Denny-Brown and Pennybacker (1938), clinical EMG rapidly developed (Buchthal & Clemmesen, 1941; Denny-Brown & Nevin, 1941; Weddell et al., 1944). In subsequent studies, Denny-Brown identified most of the clinically relevant characteristics of the EMG (see his reviews: Denny-Brown, 1949, 1953), including the interpretation of polyphasic potentials and doublets, paving the way for modern diagnostic methods and management of patients with neuromuscular disorders. More recent approaches to clinical diagnosis with EMG are mostly based on the foundations set up by Denny-Brown with the concentric needle.

## Conclusion

We now have a rather clear picture of the working principles of motor units and their activity in physiological and pathological conditions. The fundamental means through which we have reached this knowledge in nearly a century of investigations is muscle recordings, as originally shown by Adrian and Bronk in humans, which have become progressively more advanced over time. The information we extract from EMG is of utmost relevance as it represents the activity of the output layer of the somatic motor nervous system, and as such it allows us to infer the working principles of the system. Motor neurones remain today the only neural cells that we can record individually out of a pool, in humans, without surgery, in routine laboratory conditions. Moreover, with modern techniques, we can access a relatively large number of motor neurones concurrently and over long duration of time[6] (as long as the EMG activity, and therefore the

level of interference, is moderate). They constitute an ideal model of translational research from animals to humans, as Adrian and Bronk showed in their papers, as well as a target for diagnostic analysis with minimally invasive clinical procedures.

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

## Additional information

### Competing interests

No competing interests declared.

### Author contributions

Both authors have approved the final version of the manuscript and agree to be accountable for all aspects of the work. All persons designated as authors qualify for authorship, and all those who qualify for authorship are listed.

### Funding

D.F. is supported by the European Research Council (ERC) under the Synergy Grant Natural BionicS (810346) and the EPSRC Transformative Healthcare for 2050 project NISNEM

Technology (EP/T020970/1). S.G. is supported by the National Health and Medical Research Council (RG193455).

## Acknowledgements

The authors acknowledge the contribution of anonymous reviewers who helped to improve the first version of the manuscript.

## Keywords

concentric needle, electromyography, motor neurone, motor unit, muscle

## Supporting information

Additional supporting information can be found online in the Supporting Information section at the end of the HTML view of the article. Supporting information files available:

**Peer Review History**

