## [Peer Review History · The Journal of Physiology]

THE NEURAL CONTROL OF MOVEMENT: A CENTURY OF IN-VIVO MOTOR UNIT RECORDINGS IS THE LEGACY OF ADRIAN AND BRONK

Dario Farina and Simon C Gandevia

DOI: 10.1113/JP285319

Corresponding author(s): Dario Farina (d.farina@imperial.ac.uk)

The following individual(s) involved in review of this submission have agreed to reveal their identity: Oscar Garnes-Camarena (Referee #1); Andrew J. Fuglevand (Referee #2)

Review Timeline:

Submission Date:	18-Aug-2023
Editorial Decision:	25-Sep-2023
Revision Received:	10-Nov-2023
Accepted:	24-Nov-2023

Senior Editor: Laura Bennet

Reviewing Editor: Mathew Piasecki

Transaction Report:

Dear Dario,

Re: JP-TR-2023-285319 "THE NEURAL CONTROL OF MOVEMENT: A CENTURY OF IN-VIVO MOTOR UNIT RECORDINGS IS THE LEGACY OF ADRIAN AND BRONK" by Dario Farina and Simon C Gandevia

Thank you for submitting your manuscript to The Journal of Physiology. It has been assessed by a Reviewing Editor and by 3 expert referees and we are pleased to tell you that it is potentially acceptable for publication following satisfactory major revision.

REVISION CHECKLIST:

We look forward to receiving your revised submission.

Best wishes,

Professor Laura Bennet
Senior Editor
The Journal of Physiology
<https://jp.msubmit.net>
<http://jp.physoc.org>
The Physiological Society
Hodgkin Huxley House
30 Farringdon Lane
London, EC1R 3AW
UK
<http://www.physoc.org>
<http://journals.physoc.org>

REQUIRED ITEMS

-Please include an Abstract Figure file, as well as the figure legend text within the main article file. The Abstract Figure is a piece of artwork designed to give readers an immediate understanding of the Review Article and should summarise the main conclusions. If possible, the image should be easily 'readable' from left to right or top to bottom. It should show the physiological relevance of the Review so readers can assess the importance and content of the article. Abstract Figures should not merely recapitulate other figures in the Review. Please try to keep the diagram as simple as possible and without superfluous information that may distract from the main conclusion of the Review. Abstract Figures must be provided by authors no later than the revised manuscript stage and should be uploaded as a separate file during online submission labelled as File Type 'Abstract Figure'. Please ensure that you include the figure legend in the main article file. All Abstract Figures will be sent to a professional illustrator for redrawing and you may be asked to approve the redrawn figure before your paper is accepted.

-Your MS must include a complete "Additional information section" with the following 4 headings and content:

Competing Interests: A statement regarding competing interests. If there are no competing interests, a statement to this effect must be included. All authors should disclose any conflict of interest in accordance with journal policy.

Author contributions: Each author should take responsibility for a particular section of the study and have contributed to writing the paper. Acquisition of funding, administrative support or the collection of data alone does not justify authorship; these contributions to the study should be listed in the Acknowledgements. Additional information such as 'X and Y have contributed equally to this work' may be added as a footnote on the title page.

It must be stated that all authors approved the final version of the manuscript and that all persons designated as authors qualify for authorship, and all those who qualify for authorship are listed.

Funding: Authors must indicate all sources of funding, including grant numbers. If authors have not received funding, this must be stated.

It is the responsibility of authors funded by RCUK to adhere to their policy regarding funding sources and underlying research material. The policy requires funding information to be included within the acknowledgement section of a paper. Guidance on how to acknowledge funding information is provided by the Research Information Network. The policy also requires all research papers, if applicable, to include a statement on how any underlying research materials, such as data, samples or models, can be accessed. However, the policy does not require that the data must be made open. If there are considered to be good or compelling reasons to protect access to the data, for example commercial confidentiality or legitimate sensitivities around data derived from potentially identifiable human participants, these should be included in the statement.

Acknowledgements: Acknowledgements should be the minimum consistent with courtesy. The wording of acknowledgements of scientific assistance or advice must have been seen and approved by the persons concerned. This section should not include details of funding.

-Please upload separate high quality figure files via the submission form.

-Author profile(s) must be uploaded via the submission form. Authors should submit a short biography (no more than 100 words for one author or 150 words in total for two authors) and a portrait photograph of the two leading authors on the paper. These should be uploaded, clearly labelled, with the manuscript submission. Any standard image format for the photograph is acceptable, but the resolution should be at least 300 dpi and preferably more. A group photograph of all authors is also acceptable, providing the biography for the whole group does not exceed 150 words.

-It is the authors' responsibility to obtain any necessary permissions to reproduce previously published material
https://jp.msubmit.net/cgi-bin/main.plex?form_type=display_requirements#use

EDITOR COMMENTS

Reviewing Editor:

Thank you for the submission of this historical perspectives article. As you will see from the reviewer comments, the collective opinion is positive, however each reviewer has made several suggestions and requested clarity on several points which should be addressed. Where possible, I would encourage you to highlight insights garnered from animal models, and consider the points relating to the early work of Denny-Brown in relation to the size principle of MU recruitment.

Please also see 'Required Items' above.

REFEREE COMMENTS

Referee #1:

The authors present a review of Adrian and Bronk's invaluable contribution to the understanding of the physiology of the peripheral motor system. The text is well-written, timely structured and pleasant to read. Also, it is supported by a considerable number of references. Nevertheless, this reviewer would like to comment on some controversial issues in the text. The following are some clarifications and suggestions:

1. Historical note: The earliest historical reference to the text dates back to 1925, when Sherrington introduced the concept of "motor unit." He and Adrian were shortly afterwards awarded the Nobel Prize "for their discoveries on the functions of neurons". Since function is intimately related to structure, it seems worthwhile to go back a few years in time and mention the qualitative paradigm shift brought about by the neuron theory, which served as a theoretical basis for Sherrington's ideas. This researcher was directly influenced by Cajal, who was awarded the Nobel Prize in 1906 in recognition of his work on the structure of the nervous system.
2. On two occasions "muscular unit" appears in the text. The authors should be requested to clarify whether they referred to "motor unit".
3. Size Principle: There is no reference in the text to Denny-Brown's contribution to the concept of the 'Size Principle'. The authors may probably be aware of an influential review written by Mayer in 2001 on Denny-Brown's work, which argues that the initial findings and the concept underlying the size principle were originally described by Denny-Brown and Pennybacker back in 1938, although the term was later coined by Henneman. Indeed, in the original paper by Denny-Brown and Pennybacker [1938, p. 324] we can read: "The early motor units in normal gradual voluntary contraction are always in our experience small ones. The larger and more powerful motor units, each controlling many more muscle fibres, enter contraction late. In a muscle wasted by loss of many motor neurones graduation of contraction becomes uneven because large motor units become active without sufficient previous recruitment of small units". In this regard, other authors have proposed to rename the Henneman Size Principle (Vilensky and Gilman 1998).

4. Action potential: A distinction between a motor unit action potential and a motor unit potential is suggested. Although in the literature both concepts are used interchangeably, the definition of action potential accounts for a transmembrane phenomenon, whereas the latter represents the local field potential registered at a distance. An individual muscle fiber action potential can be measured with specific methods such as the patch clamp, whereas a motor unit potential is the temporal summation of the volume-conducted potentials from the muscle fibers belonging to a motor unit, and it is measured with electromyographic needles located in the territory of the motor unit.

5. On p.14 we read: "These developments have progressively allowed the full non-invasive decoding of motor neuronal activity in humans". This statement seems to indicate that by means of surface EMG the motor activity can be fully decoded. Surface EMG has several important limitations that deter its use in clinical neurophysiology (for instance, detection of fibrillation potentials or polyphasic potentials).

6. On p.15 we read: "The processing methods are now automatic algorithms or analogue real-time systems that replace visual inspection, but they are based on the discrimination of action potential waveforms of different motor units". In recent decades there have been some contributions on quantitative EMG not based on discrimination of motor unit potentials. Authors are advised to review the work of Nandedkar (1986a & 1986b) on interference pattern analysis and Fuglsang-Frederiksen on peak ratio (1985), that describe methods included in commercially available EMG machines for clinical use.

7. On p.21: "The concentric needle is still used in clinical neurophysiology examinations in medical centres globally to diagnose neuromuscular disorders, such as dystrophies". The text would be more complete if some examples of the multiple diagnostic applications of needle EMG were added. Certainly, dystrophies are included in the panoply of neuromuscular disorders, yet their incidence is significantly lower than other neurogenic and myopathic disorders for which needle EMG could be used.

8. On P.23 we read: "The information we extract from EMG is of utmost relevance as it represents the activity of the output layer of the nervous system". EMG represents the output of the somatic efferent nervous system. A clarification that the text is referring to the somatic motor nervous system is required, for autonomic neurons are also efferent, yet not suitably explored with EMG techniques.

9. Also on p.23: "Motor neurones remain today the only neural cells from which we can record in humans without surgery, in routine laboratory conditions". I would suggest a clarification of this statement. Do the authors mean individual recordings? During nerve conduction studies (NCS), under laboratory conditions, the electrical activity of groups of motor and sensory neurons can be recorded. What makes motor neurons unique is the fact that the electrical activity of each of them can be studied individually from a pool of active motor neurons, as long as the level of activation is low to moderate (further increases in the activation levels interfere with the identification of individual potentials, hence the name interference pattern).

10. Finally, some of the references listed have an uneven format (e.g., the initial of the name is not followed by a period, the name of the journal is not followed by a colon, some references include the DOI code while other refer to the PMID).

Referee #2:

General Comments

This was a very interesting, important, and welcome account of the pioneering work of Adrian and Bronk related to motor unit recording in humans. The manuscript was very nicely written - smoothly weaving together the history of this method with current approaches. This will certainly be a widely read contribution by two of the most important investigators of human motor neuron function.

Major Recommendations

None.

Minor Recommendations

1. The manuscript might benefit from a few subheadings.

2. (pg 3, para 2, also pg 12, last line) "described the output of the spinal cord during autonomous, reflex, and voluntary" - It

may be obvious, but I am uncertain as to what is meant by "autonomous" in the context of present day motor control. Is it meant to refer to movements mediated by central pattern generators, postural control, innate behaviors? While that might have been the term of use at the time of Adrian and Bronk's publication, some explanation could be useful.

3. (pg 5, para 1) "well as their proposal to study nerve activity by recording signals from human muscles." While at the time, Adrian and Bronk may have emphasized the utility of studying "nerve" activity by recording from muscle, what seems more salient is that such recording enables recording of motor neuron activity.

4. In the caption for Figure 1, please briefly describe what the symbols "a", "a'", "x" and "b", "b'", and "x'"

5. (pg 6, para 1) "Recordings of the activity of single nerve fibres were performed with a dissection technique described" Please indicate what creature was used for these experiments.

6. For the caption of Figure 2c, please indicate that the display is such that action potentials are represented as downward deflections of the white signal. Also, mention the timing marks and what was the time between marks.

7. (pg 8, para 2) "related to the duration of single fibre discharges" This phrase could be misinterpreted to mean the duration of single action potentials, rather than the time span over which a single unit was active.

8. (pg 9, para 3) "Along this line, smooth muscle" - this should be re-written because it seems as though you are referring to smooth muscle rather than striated muscle.

9. (pg 10, para 1) "Here they chose several nerves for dissection, mainly in the lower limb" - should point out in what animal.

10. Figure 3a - should point out these recordings were done in the cat.

11. Figure 3b needs some further explanation. At first glance, given the semicircular profile of the firing rate vs time plot in Figure 3a, one might think (as I did!) that the plots in 3b are schematic representations of such activity. However, I don't believe the horizontal axis in 3b represents time (not sure what it is). Nevertheless, it seems that the semicircles in 3b are like the sun rising on the horizon, and the further it rises, the more units are recruited. The units in the center represent the earliest recruited units, and those more peripheral are later recruited. Not sure what the shading represents in the lower part of Figure 3b.

12. (pg 16, para 2) "the size principle and was described by Henneman approximately 30 years after the papers by Adrian and Bronk (Henneman, 1957) (Figure 4). This principle is based on biophysical properties of the motor neurones, i.e. on the difference in impedance between small and large motor neurones that translates to a difference in threshold of excitability" While this may not be the venue for this matter (I leave this up to the authors), the concept of input resistance, cell size, and susceptibility to excitation was originally articulated by Katz and Thesleff in the *Journal of Physiology* (1957) based on studies of muscle fibers of different diameters. Not to take away credit from his seminal findings, but Henneman offered no mechanistic explanation for the greater susceptibility of neurons with thin axons to be activated in his 1957 paper. Later, however, Henneman et al. (1965) cite Katz and Thesleff (1957) for providing this crucial idea.

13. (pg 21, ln 1) "following exercise training (Christie & Kamen, 2010; Van Cutsem et al., 1998; Duchateau et al., 2005)" The authors humbly neglected to cite one of the most important papers along these lines from the first author's own laboratory: Del Vecchio et al. *J Physiol*, 2019.

14. (pg 21, para) "Clinical EMG by needle electrodes currently remains by far the most relevant application among the (invasive and non-invasive) EMG techniques." Perhaps re-write as:

"EMG using needle electrodes currently remains by far the most relevant clinical application..."

15. Figure 5 could use some slight changes or additions. For example, the legend to Fig. 5a states "When recording the discharge times of pairs of motor neurons", yet the discharge of only a single neuron is shown in subpanel A. The threshold crossing diagrams in subpanels B and C do not seem directly relevant to the matter at hand. Perhaps all that is needed in Figure 5a is the schematic shown in subpanel D. In Fig. 5b, it does not seem necessary to include both cross-correlograms in the context of the present manuscript.

16. (pg 23, para 1) "Motor neurones remain today the only neural cells from which we can record in humans without surgery, in routine laboratory conditions" With microneurographic methods, one can also record from dorsal root ganglion neurons in humans without surgery.

Referee #3:

Please see the attached file.

END OF COMMENTS

Confidential Review

18-Aug-2023

This is an interesting, instructive and important historical perspective which is generally very readable and informative and that should have broad appeal and interest. I share some comments and questions for perhaps improved clarity, comprehensiveness and accuracy that I trust the authors will take under consideration. I thank the authors for taking the time and interest in creating this excellent perspective.

- 1) Would it be helpful to include a very brief biography of who are these people (Adrian and Bronk), where did they 'come from', do their work, their relationship and did they continue to work in this area after the seminal papers in 1928/29. I believe readers would find some high level details of interest early in the manuscript.
- 2) Not to potentially undermine these important early studies of Adrian and Bronk, but were these types of recording also made earlier or at approximately the same time by a German lab (Wachholder K. Willkürliche Haltung und Bewegung. *Ergeb Physiol* 568–775, 1928.)? Is this helpful to note if true?
- 3) My version did not have line numbers unfortunately – 2nd para on page 3, 7th line – that sentence or phrasing beginning with "... comparable data as from direct nerve fibre recordings by recording indirectly from muscles." Is very awkward and could be improved for readability and clarity.
- 4) Top of page 10 – first 4 lines: Using the Macefield reference to simply state that these two assumptions have now been shown not to be true seems perplexing. In that paper in 1996 those authors used microneurography to stimulate individual axons whereas the assumptions seem to pertain to voluntary contractions. Is this a reasonable or fair comparison? It might be helpful if so to give some brief further explanation of how/why these assumptions are incorrect rather than simply a presumed exemplar reference that on the surface does not seem to address this issue per se.
- 5) End page 11 to top of page 12 – I am unsure of the value of this aside about muscles related to breathing and seems a bit odd to suggest that had Adrian used this muscle everything would have been thrown off.
- 6) Top paragraph of page 14 – Have researchers not also used essentially single monopolar needles to record from muscles as well as these others noted – following along the Stalberg single fibre monopolar electrode idea into the world of microneurography initiated by several people (Hagbarth, Vallbo, Wallin and Johansson) for nerve recordings, but also used for motor unit recordings in muscle (Bigland-Ritchie and others). Should that type of electrode also be recognised to be inclusive to the field?
- 7) Did Denny-Brown not basically describe the size principle years before Henneman?

- 8) Is there any updated histological evidence to support common drive wiring? Do I recall somewhere seeing that common drive in NHP can be bypassed to select single MUs for recruitment that are under cortical control. Again for completeness if true this may be valuable for the readership; unless it seems too specific and outside the particular framework here.
- 9) Page 17, 3 lines up from end of page – does the use of the word ‘failed’ seem a little harsh. Perhaps simply ‘did not’.
- 10) Page 20 , 4 lines from end using the Fuglevand citation to suggest that excitability of MNs may differ in MNs of different sizes. While this may be true, that paper discussing saturation tested units at less than 10% MVC and thus it seems an unlikely citation for this idea as presumably only small low threshold units were tested in that study and it would not address the concept of different sized MNs. Perhaps some Heckman or others’ papers might be more pertinent to support this statement.
- 11) It is stated throughout this review that recording from muscle fibres as the end organ output of the motor pathway is directly reflecting the discharges of the MN. Has this been directly tested/compared in any model – do we know for sure that what is being recorded in the muscle exactly reflects the MN output with a faithful one to one signal response? It would be helpful if a brief mention of experiments that have clearly substantiated this have been reported. This may be more important of course when studying known pathologies that might affect the proper workings of the NMJ, axonal conductance and so forth. Despite what is being sent forth from the soma of MN, may not be what is being ‘received’ by the muscle fibres when recording using EMG. Indeed this factor may be related to normal ageing that the fidelity of the MN output may not be faithfully and completely delivered to the muscle fibres because of age-related loss of safety factors or NMJ architectural changes and perhaps other factors. We assume that in a healthy young adult/animal that this relationship is solid, but this point perhaps should be substantiated for the reader (if it has been comprehensively validated) or noted as a limitation/assumption and especially when recording EMG in a muscle with certain NM pathologies. The window into MN function from the muscle may not always be transparent.

EDITOR COMMENTS

Reviewing Editor:

Thank you for the submission of this historical perspectives article. As you will see from the reviewer comments, the collective opinion is positive, however each reviewer has made several suggestions and requested clarity on several points which should be addressed. Where possible, I would encourage you to highlight insights garnered from animal models, and consider the points relating to the early work of Denny-Brown in relation to the size principle of MU recruitment.

REPLY:

We thank the reviewer and Editors for very useful comments that have substantially helped us to improve the quality and accuracy of this work. We have addressed each comment, with corresponding changes in the manuscript. Point-by-point replies with explanations on how we addressed the comments follow.

Please also see 'Required Items' above.

REFEREE COMMENTS

Referee #1:

The authors present a review of Adrian and Bronk's invaluable contribution to the understanding of the physiology of the peripheral motor system. The text is well-written, timely structured and pleasant to read. Also, it is supported by a considerable number of references. Nevertheless, this reviewer would like to comment on some controversial issues in the text. The following are some clarifications and suggestions:

1. Historical note: The earliest historical reference to the text dates back to 1925, when Sherrington introduced the concept of "motor unit." He and Adrian were shortly afterwards awarded the Nobel Prize "for their discoveries on the functions of neurons". Since function is intimately related to structure, it seems worthwhile to go back a few years in time and mention the qualitative paradigm shift brought about by the neuron theory, which served as a theoretical basis for Sherrington's ideas. This researcher was directly influenced by Cajal, who was awarded the Nobel Prize in 1906 in recognition of his work on the structure of the nervous system.

REPLY: Thank you for this suggestion. We have added a reference to Cajal's work and a sentence indicating its influence on Sherrington.

2. On two occasions "muscular unit" appears in the text. The authors should be requested to clarify whether they referred to "motor unit".

REPLY: With muscle unit we intend the group of muscle fibres innervated by a single motor neurone, and therefore to a part of the motor unit. This was defined in the original version of the manuscript, but we have now revised that part of the text to make the definition clearer.

3. Size Principle: There is no reference in the text to Denny-Brown's contribution to the concept of the 'Size Principle'. The authors may probably be aware of an influential review written by Mayer in 2001 on Denny-Brown's work, which argues that the initial findings and the concept underlying the size principle were originally described by Denny-Brown and Pennybacker back in 1938, although the term was later coined by Henneman. Indeed, in the original paper by Denny-Brown and Pennybacker [1938, p. 324] we can read: "The early motor units in normal gradual voluntary contraction are always in our experience small ones. The larger and more powerful motor units, each controlling many more muscle fibres, enter contraction late. In a muscle wasted by loss of many motor neurones graduation of contraction becomes uneven because large motor units become active without sufficient previous recruitment of small units". In this regard, other authors have proposed to rename the Henneman Size Principle (Vilensky and Gilman 1998).

REPLY: We have now added a more complete historical perspective of the size principle. Despite the early observation by Denny-Brown and Pennybacker, we are in line with the opinion expressed by Fuglevand in his reply letter to the letter by Vilensky and Gilman. While it is correct that Denny-Brown and Pennybacker reported an example of voluntary recording from the human biceps brachii muscle and commented on the progressive increase in size of the recruited motor units, it was Henneman who first proposed the neural mechanisms underlying such observations and that made it a true universal principle applicable to all types of neurons. We now report more references in this respect and provide an overview of the different opinions on the paternity of the size principle.

4. Action potential: A distinction between a motor unit action potential and a motor unit potential is suggested. Although in the literature both concepts are used interchangeably, the definition of action potential accounts for a transmembrane phenomenon, whereas the latter represents the local field potential registered at a distance. An individual muscle fiber action potential can be measured with specific methods such as the patch clamp, whereas a motor unit potential is the temporal summation of the volume-conducted potentials from the muscle fibers belonging to a motor unit, and it is measured with electromyographic needles located in the territory of the motor unit.

REPLY: This is a fine distinction on which we agree and we thank the reviewer for bringing this point to our attention. As said, the distinction is usually not applied in the literature where the cited terms are used interchangeably. Nonetheless, we appreciate the suggestion and are happy to follow it (this has been adjusted in the entire manuscript).

5. On p.14 we read: "These developments have progressively allowed the full non-invasive decoding of motor neuronal activity in humans". This statement seems to indicate that by means of surface EMG the motor activity can be fully decoded. Surface EMG has several important limitations that deter its use in clinical neurophysiology (for instance, detection of fibrillation potentials or polyphasic potentials).

REPLY: We agree fully and have modified the cited sentence accordingly.

6. On p.15 we read: "The processing methods are now automatic algorithms or analogue real-time systems that replace visual inspection, but they are based on the discrimination of action potential

waveforms of different motor units". In recent decades there have been some contributions on quantitative EMG not based on discrimination of motor unit potentials. Authors are advised to review the work of Nandedkar (1986a & 1986b) on interference pattern analysis and Fuglsang-Frederiksen on peak ratio (1985), that describe methods included in commercially available EMG machines for clinical use.

REPLY: We have now mentioned these methods and added references.

7. On p.21: "The concentric needle is still used in clinical neurophysiology examinations in medical centres globally to diagnose neuromuscular disorders, such as dystrophies". The text would be more complete if some examples of the multiple diagnostic applications of needle EMG were added. Certainly, dystrophies are included in the panoply of neuromuscular disorders, yet their incidence is significantly lower than other neurogenic and myopathic disorders for which needle EMG could be used.

REPLY: We have substituted dystrophies with a more specific and commonly diagnosed disorder with needle EMG.

8. On P.23 we read: "The information we extract from EMG is of utmost relevance as it represents the activity of the output layer of the nervous system". EMG represents the output of the somatic efferent nervous system. A clarification that the text is referring to the somatic motor nervous system is required, for autonomic neurons are also efferent, yet not suitably explored with EMG techniques.

REPLY: Corrected, thanks.

9. Also on p.23: "Motor neurones remain today the only neural cells from which we can record in humans without surgery, in routine laboratory conditions". I would suggest a clarification of this statement. Do the authors mean individual recordings? During nerve conduction studies (NCS), under laboratory conditions, the electrical activity of groups of motor and sensory neurons can be recorded. What makes motor neurons unique is the fact that the electrical activity of each of them can be studied individually from a pool of active motor neurons, as long as the level of activation is low to moderate (further increases in the activation levels interfere with the identification of individual potentials, hence the name interference pattern).

REPLY: Corrected, as suggested.

10. Finally, some of the references listed have an uneven format (e.g., the initial of the name is not followed by a period, the name of the journal is not followed by a colon, some references include the DOI code while other refer to the PMID).

REPLY: We have now made out best to uniform all references.

Referee #2:

General Comments

This was a very interesting, important, and welcome account of the pioneering work of Adrian and Bronk related to motor unit recording in humans. The manuscript was very nicely written - smoothly weaving together the history of this method with current approaches. This will certainly be a widely read contribution by two of the most important investigators of human motor neuron function.

REPLY: We thank the reviewer for the appreciation of the work.

Major Recommendations

None.

Minor Recommendations

1. The manuscript might benefit from a few subheadings.

REPLY: We have considered and tried to follow this suggestion. In the end, we would prefer to keep the structure as it is, with only main headings. This is because the style we have used is of an historical narrative where each section contains interlinked paragraphs. We believe adding subheadings would fragment the main narrative flow. We hope this is acceptable.

2. (pg 3, para 2, also pg 12, last line) "described the output of the spinal cord during autonomous, reflex, and voluntary" - It may be obvious, but I am uncertain as to what is meant by "autonomous" in the context of present day motor control. Is it meant to refer to movements mediated by central pattern generators, postural control, innate behaviors? While that might have been the term of use at the time of Adrian and Bronk's publication, some explanation could be useful.

REPLY: The autonomic nervous system (sometimes also called involuntary nervous system) regulates the involuntary functions of the body, such as heart rate, blood pressure, digestion, and sweating. In the case of Adrian and Bronk, they recorded from the phrenic nerve that provides input to the diaphragm, the primary muscle for inspiration. This is what we refer to as autonomous contractions. We have now specified in the revised text, respiration as the specific contractions considered.

3. (pg 5, para 1) "well as their proposal to study nerve activity by recording signals from human muscles." While at the time, Adrian and Bronk may have emphasized the utility of studying "nerve" activity by recording from muscle, what seems more salient is that such recording enables recording of motor neuron activity.

REPLY: We agree. We have now mentioned motor neuron activity in the cited sentence.

4. In the caption for Figure 1, please briefly describe what the symbols "a", "a'", "x" and "b", "b'", and "x'"

REPLY: Done.

5. (pg 6, para 1) "Recordings of the activity of single nerve fibres were performed with a dissection technique described" Please indicate what creature was used for these experiments.

REPLY: Done (rabbit for the first paper).

6. For the caption of Figure 2c, please indicate that the display is such that action potentials are represented as downward deflections of the white signal. Also, mention the timing marks and what was the time between marks.

REPLY: The figure reports an oscillatory wave (sinusoid) generated by the oscilloscope. The caption and the text do not mention a time reference for this waveform but, given the numbers provided for the firing rate in the different plots, the distance between peaks of the sinusoidal waveform is estimated as 10 ms. We now report this as an estimated value in the caption.

7. (pg 8, para 2) "related to the duration of single fibre discharges" This phrase could be misinterpreted to mean the duration of single action potentials, rather than the time span over which a single unit was active.

REPLY: Corrected to avoid ambiguity.

8. (pg 9, para 3) "Along this line, smooth muscle" - this should be re-written because it seems as though you are referring to smooth muscle rather than striated muscle.

REPLY: Rephrased to avoid ambiguity.

9. (pg 10, para 1) "Here they chose several nerves for dissection, mainly in the lower limb" - should point out in what animal.

REPLY: Specified (cat, in the second paper).

10. Figure 3a - should point out these recordings were done in the cat.

REPLY: Done (in the caption).

11. Figure 3b needs some further explanation. At first glance, given the semicircular profile of the firing rate vs time plot in Figure 3a, one might think (as I did!) that the plots in 3b are schematic representations of such activity. However, I don't believe the horizontal axis in 3b represents time (not sure what it is). Nevertheless, it seems that the semicircles in 3b are like the sun rising on the horizon, and the further it rises, the more units are recruited. The units in the center represent the earliest recruited units, and those more peripheral are later recruited. Not sure what the shading represents in the lower part of Figure 3b.

REPLY: We have added a more extensive explanation in the captions of this figure, which may indeed not be crystal clear.

12. (pg 16, para 2) "the size principle and was described by Henneman approximately 30 years after the papers by Adrian and Bronk (Henneman, 1957) (Figure 4). This principle is based on biophysical properties of the motor neurones, i.e. on the difference in impedance between small and large motor neurones that translates to a difference in threshold of excitability" While this may not be the

venue for this matter (I leave this up to the authors), the concept of input resistance, cell size, and susceptibility to excitation was originally articulated by Katz and Thesleff in the Journal of Physiology (1957) based on studies of muscle fibers of different diameters. Not to take away credit from his seminal findings, but Henneman offered no mechanistic explanation for the greater susceptibility of neurons with thin axons to be activated in his 1957 paper. Later, however, Henneman et al. (1965) cite Katz and Thesleff (1957) for providing this crucial idea.

REPLY: Thank you for this insightful comment. We have revised substantially the part related to the size principle, also in relation to comments by other reviewers. We have now cited the work by Katz and Thesleff as well as contributions to the identification of the size principle by Denny-Brown. We believe this section on size principle is now much better balanced.

13. (pg 21, ln 1) "following exercise training (Christie & Kamen, 2010; Van Cutsem et al., 1998; Duchateau et al., 2005)" The authors humbly neglected to cite one of the most important papers along these lines from the first author's own laboratory: Del Vecchio et al. J Physiol, 2019.

REPLY: Thank you for the kind comment. We have now added this reference.

14. (pg 21, para) "Clinical EMG by needle electrodes currently remains by far the most relevant application among the (invasive and non-invasive) EMG techniques." Perhaps re-write as:

"EMG using needle electrodes currently remains by far the most relevant clinical application..."

REPLY: We have now rephrased but in a different way than proposed by the reviewer. We believe the sentence is clearer now.

15. Figure 5 could use some slight changes or additions. For example, the legend to Fig. 5a states "When recording the discharge times of pairs of motor neurons", yet the discharge of only a single neuron is shown in subpanel A. The threshold crossing diagrams in subpanels B and C do not seem directly relevant to the matter at hand. Perhaps all that is needed in Figure 5a is the schematic shown in subpanel D. In Fig. 5b, it does not seem necessary to include both cross-correlograms in the context of the present manuscript.

REPLY: We agree. We have now reduced Figure 5 to the essential elements, eliminating the parts not needed, as suggested, and have extended the caption to add explanations.

16. (pg 23, para 1) "Motor neurones remain today the only neural cells from which we can record in humans without surgery, in routine laboratory conditions" With microneurographic methods, one can also record from dorsal root ganglion neurons in humans without surgery.

REPLY: We thank for this observation. We have added a footnote to mention this important point.

Referee #3:

This is an interesting, instructive and important historical perspective which is generally very readable and informative and that should have broad appeal and interest. I share some comments and questions for perhaps improved clarity, comprehensiveness and accuracy that I trust the authors will take under consideration. I thank the authors for taking the time and interest in creating this excellent perspective.

REPLY: We are grateful for the appreciation of our work.

1) Would it be helpful to include a very brief biography of who are these people (Adrian and Bronk), where did they 'come from', do their work, their relationship and did they continue to work in this area after the seminal papers in 1928/29. I believe readers would find some high level details of interest early in the manuscript.

REPLY: We have preferred not to add a separate section with the biographies of the two original authors. Nonetheless, to meet the suggestion of the reviewer, we have added substantial details on their life, including what they did after their seminal papers (see footnote on Bronk, for example). This is done in a way that is integrated in the text.

2) Not to potentially undermine these important early studies of Adrian and Bronk, but were these types of recording also made earlier or at approximately the same time by a German lab (Wachholder K. Willkürliche Haltung und Bewegung. *Ergeb Physiol* 568–775, 1928.)? Is this helpful to note if true?

REPLY: Thank you for this reference. We have found it and translated as much as we could. This German laboratory was indeed doing muscle recordings with needle electrodes (similar needle electrodes are described earlier in "Rehn, E. Elektrophysiologie krankhaft veränderter menschlicher Muskeln. *Deutsche Zeitschrift f. Chirurgie* 162, 155–167 (1921)"). These needle electrodes were made of steel (sewing needles) as well as nickel silver and recorded muscle signals with various degrees of selectivity. The cited reference is very interesting in discussing how needle recordings are needed to distinguish specific electrical activity of muscles. In some of their figures, it is also possible to recognise what are likely single motor unit action potentials (e.g., their Figure 10). Yet, they do not interpret them as single motor unit activities and provide an explanation of these activities as a mixture of interference patterns and waveforms changing amplitude over time (see their explanatory Figure 11). While we agree that these results should be cited, we argue that they do not diminish the original contribution by Adrian & Bronk. We have added the reference and comments on it and we thank the reviewer for such helpful information.

3) My version did not have line numbers unfortunately – 2nd para on page 3, 7th line – that sentence or phrasing beginning with "... comparable data as from direct nerve fibre recordings by recording indirectly from muscles." Is very awkward and could be improved for readability and clarity.

REPLY: Rephrased.

4) Top of page 10 – first 4 lines: Using the Macefield reference to simply state that these two assumptions have now been shown not to be true seems perplexing. In that paper in 1996 those authors used microneurography to stimulate individual axons whereas the assumptions seem to pertain to voluntary contractions. Is this a reasonable or fair comparison? It might be helpful if so to give some brief further explanation of how/why these assumptions are incorrect rather than simply a presumed exemplar reference that on the surface does not seem to address this issue per se.

REPLY: We have followed the suggestion and provided a brief explanation of the statement, also adding a reference.

5) End page 11 to top of page 12 – I am unsure of the value of this aside about muscles related to breathing and seems a bit odd to suggest that had Adrian used this muscle everything would have been thrown off.

REPLY: We agree. We have added one sentence introducing this part and, especially, we have moved it to a footnote. As noted by the reviewer, the logic flow of reasoning of the main text seemed fragmented by this statement.

6) Top paragraph of page 14 – Have researchers not also used essentially single monopolar needles to record from muscles as well as these others noted – following along the Stalberg single fibre monopolar electrode idea into the world of microneurography initiated by several people (Hagbarth, Vallbo, Wallin and Johannson) for nerve recordings, but also used for motor unit recordings in muscle (Bigland-Ritchie and others). Should that type of electrode also be recognised to be inclusive to the field?

REPLY: Thank you for the note. We have now included the tungsten electrode developed for microneurography and later used by Bigland-Ritchie.

7) Did Denny-Brown not basically describe the size principle years before Henneman?

REPLY: We have largely re-written the part on size principle to recognise the contributions of Denny-Brown and others before Henneman's. In doing so, we have provided additional references. This part is much more balanced than in the original version.

8) Is there any updated histological evidence to support common drive wiring? Do I recall somewhere seeing that common drive in NHP can be bypassed to select single MUs for recruitment that are under cortical control. Again for completeness if true this may be valuable for the readership; unless it seems too specific and outside the particular framework here.

REPLY: A recent study by Marshall et al. (Flexible neural control of motor units. *Nat Neurosci.* 2022 Nov; 25(11): 1492–1504) has indeed proposed the possibility of flexible control of motor units, basically opposite to a common input control. These data have been discussed extensively (for our view, see Hug et al. 2023 in the reference list of the current manuscript) but their detailed discussion is too specific for this contribution. We respectfully prefer to avoid it in this work.

9) Page 17, 3 lines up from end of page – does the use of the word 'failed' seem a little harsh. Perhaps simply 'did not'.

REPLY: Agree. It has been changed as suggested.

10) Page 20, 4 lines from end using the Fuglevand citation to suggest that excitability of MNs may differ in MNs of different sizes. While this may be true, that paper discussing saturation tested units at less than 10% MVC and thus it seems an unlikely citation for this idea as presumably only small low threshold units were tested in that study and it would not address the concept of different sized MNs. Perhaps some Heckman or others' papers might be more pertinent to support this statement.

REPLY: We have associated the first part of the sentence to the reference by Fuglevand and added a new reference for the second part of the sentence.

11) It is stated throughout this review that recording from muscle fibres as the end organ output of the motor pathway is directly reflecting the discharges of the MN. Has this been directly tested/compared in any model – do we know for sure that what is being recorded in the muscle exactly reflects the MN output with a faithful one to one signal response? It would be helpful if a brief mention of experiments that have clearly substantiated this have been reported. This may be more important of course when studying known pathologies that might affect the proper workings of the NMJ, axonal conductance and so forth. Despite what is being sent forth from the soma of MN, may not be what is being 'received' by the muscle fibres when recording using EMG. Indeed this

factor may be related to normal ageing that the fidelity of the MN output may not be faithfully and completely delivered to the muscle fibres because of age-related loss of safety factors or NMJ architectural changes and perhaps other factors. We assume that in a healthy young adult/animal that this relationship is solid, but this point perhaps should be substantiated for the reader (if it has been comprehensively validated) or noted as a limitation/assumption and especially when recording EMG in a muscle with certain NM pathologies. The window into MN function from the muscle may not always be transparent.

REPLY: Thank you for this note. We have now added a footnote to comment on this, including a reference that shows potential conduction block in pathology.

Dear Dario,

Re: JP-TR-2023-285319R1 "THE NEURAL CONTROL OF MOVEMENT: A CENTURY OF IN-VIVO MOTOR UNIT RECORDINGS IS THE LEGACY OF ADRIAN AND BRONK" by Dario Farina and Simon C Gandevia

We are pleased to tell you that your paper has been accepted for publication in The Journal of Physiology.

Please see comments below for one small change that should be made at proof stage.

Authors should note that it is too late at this point to offer corrections prior to proofing. Major corrections at proof stage, such as changes to figures, will be referred to the Editors for approval before they can be incorporated. Only minor changes, such as to style and consistency, should be made at proof stage. Changes that need to be made after proof stage will usually require a formal correction notice.

Best wishes,

Professor Laura Bennet
Senior Editor
The Journal of Physiology
<https://jp.msubmit.net>
<http://jp.physoc.org>
The Physiological Society
Hodgkin Huxley House
30 Farringdon Lane
London, EC1R 3AW
UK
<http://www.physoc.org>
<http://journals.physoc.org>

P.S. - You can help your research get the attention it deserves! Check out Wiley's free Promotion Guide for best-practice recommendations for promoting your work at www.wileyauthors.com/eeo/guide. You can learn more about Wiley Editing Services which offers professional video, design, and writing services to create shareable video abstracts, infographics, conference posters, lay summaries, and research news stories for your research at www.wileyauthors.com/eeo/promotion.

IMPORTANT NOTICE ABOUT OPEN ACCESS: To assist authors whose funding agencies mandate public access to published research findings sooner than 12 months after publication, The Journal of Physiology allows authors to pay an Open Access (OA) fee to have their papers made freely available immediately on publication.

You can check if your funder or institution has a Wiley Open Access Account here: <https://authorservices.wiley.com/author-resources/Journal-Authors/licensing-and-open-access/open-access/author-compliance-tool.html>.

EDITOR COMMENTS

Reviewing Editor:

Thank you for providing such a comprehensive response. The collective opinion is very positive and I look forward to seeing

this published.

Regarding the final point from reviewer #1, please amend point #5, p.17. Perhaps something along the lines of "These developments have progressively allowed for the full non-invasive decoding of voluntary motor neurone activity in superficial muscles of humans", or something similar. Please note, this can be amended at proofing stage.

REFEREE COMMENTS

Referee #1:

I acknowledge the effort made by the authors to include the suggested comments in the text and congratulate them on the result. In my opinion, the manuscript has improved markedly.

However, we can still read in the text (p.17) that "these developments [i.e., surface EMG] have progressively allowed for the full non-invasive decoding of motor neurone activity in humans", a sentence that the authors agreed to modify. I suggest rephrasing it, given the excessive enthusiasm it conveys, in contrast to the current limited clinical use of surface EMG.

Other than this, I have no further objections.

Referee #2:

The authors have carefully addressed the minor concerns I had with the original manuscript.

Referee #3:

Thank you for your considered responses and modifications to my queries and suggestions. The manuscript is in good shape and I have no further comments. I look forward to seeing it published.

1st Confidential Review

10-Nov-2023